# Prediction Model of Flavonoids Content in Ancient Tree Sun−Dried Green Tea under Abiotic Stress Based on LASSO−Cox

Lei Li [1,†], Yamin Wu [1,†], Houqiao Wang [1], Junjie He [1], Qiaomei Wang [1], Jiayi Xu [1], Yuxin Xia [2], Wenxia Yuan [1], Shuyi Chen [1], Lin Tao [3], Xinghua Wang [1,*] and Baijuan Wang [1,*]

[1] College of Tea Science, Yunnan Agricultural University, Kunming 650201, China; 18387047405@163.com (L.L.); m1678447405@163.com (Y.W.); 17787464434@163.com (H.W.); junjie19991207@163.com (J.H.); wqm19850127@163.com (Q.W.); 18088111511@163.com (J.X.); yuanwenxia2023@163.com (W.Y.); 19184037854@163.com (S.C.)

[2] College of Mechanical and Electrical Engineering, Yunnan Agricultural University, Kunming 650201, China; 15140964046@163.com

[3] Pu'er Wenbang Tea Co., Ltd., Pu'er 666500, China; 18987888396@163.com

[*] Correspondence: wangaoyu2011@163.com (X.W.); wangbaijuan2023@163.com (B.W.)

[†] These authors contributed equally to this work.

**Abstract:** To investigate the variation in flavonoids content in ancient tree sun–dried green tea under abiotic stress environmental conditions, this study determined the flavonoids content in ancient tree sun−dried green tea and analyzed its correlation with corresponding factors such as the age, height, altitude, and soil composition of the tree. This study uses two machine−learning models, Least Absolute Shrinkage and Selection Operator (LASSO) regression and Cox regression, to build a predictive model based on the selection of effective variables. During the process, bootstrap was used to expand the dataset for single−factor and multi−factor comparative analyses, as well as for model validation, and the goodness−of−fit was assessed using the Akaike information criterion (*AIC*). The results showed that pH, total potassium, nitrate nitrogen, available phosphorus, hydrolytic nitrogen, and ammonium nitrogen have a high accuracy in predicting the flavonoids content of this model and have a synergistic effect on the production of flavonoids in the ancient tree tea. In this prediction model, when the flavonoids content was >6‰, the area under the curve of the training set and validation set were 0.8121 and 0.792 and, when the flavonoids content was >9‰, the area under the curve of the training set and validation set were 0.877 and 0.889, demonstrating good consistency. Compared to modeling with all significantly correlated factors (*p* < 0.05), the *AIC* decreased by 32.534%. Simultaneously, a visualization system for predicting flavonoids content in ancient tree sun−dried green tea was developed based on a nomogram model. The model was externally validated using actual measurement data and achieved an accuracy rate of 83.33%. Therefore, this study offers a scientific theoretical foundation for explaining the forecast and interference of the quality of ancient tree sun−dried green tea under abiotic stress.

**Keywords:** flavonoids content; LASSO; nomogram; Cox regression; prediction model

## 1. Introduction

Ancient tea trees refer to wild tea trees and their communities that are distributed in natural forests, semi−domesticated artificially cultivated wild tea trees, and ancient tea gardens (forests) that have been cultivated for over 100 years. They hold high economic value and scientific research value. The green tea processed from fresh leaves of ancient tea trees through the steps of fixation, rolling, and sun−drying is called ancient tree sun–dried green tea. Made from sun−dried ancient tea trees, raw green tea is highly acclaimed in the market for its harmonious taste, great steeping endurance, and long–lasting sweet aftertaste. Flavonoids have a soft astringency and bitterness on the taste of ancient tree sun−dried

green tea [1,2], which is an important component of green tea soup color. Tea is recognized as the optimal source of nutritional flavonoids, which the human body is incapable of synthesizing [3]; it has significant effects on human body, including enzyme regulation, improvement of blood circulation, regulation of cell apoptosis, prevention of abnormal fat accumulation, anticancer properties, inhibition of periodontal disease, antioxidant properties, weight loss, blood sugar reduction, and attenuation of neurodegeneration, such as Parkinson's disease (PD), Alzheimer's disease (AD), etc. [4–13]. In addition, research has shown that total flavonoids can serve as an innovative versatile electrolyte additive to enhance the cycling efficiency of high–voltage $LiNiMnCoO_2$ lithiumion batteries [14].

The content of total flavonoids in tea trees is determined by factors such as tea tree variety, ecological environment, soil condition, and management methods. Wang et al. [15] employed a combination of principal component analysis (PCA) and artificial neural networks (ANN) to establish prediction models for buckwheat starch, protein, and total flavonoids content, respectively. The results revealed low accuracy in predicting protein and total flavonoids content, which may be due to the limited sample size, and no appropriate methods were applied to address this problem. Kusumiyati et al. [16] used near–infrared spectroscopy (NIRS) combined with machine learning to predict the total phenolic content (TPC) and total flavonoids content (TFC) in several horticultural products and attained positive outcomes. Nonetheless, currently, there is no available prediction of flavonoids content in ancient tree sun−dried green tea.

The present study selects ancient tree sun–dried green tea from Laowu Mountain as the research subjects and aims to explore the contribution of diverse environmental factors (for instance, altitude [17], soil pH [18], organic matter [19], and potassium content [20]) on the flavonoids content in sun–dried green tea. Through the construction of a multiple regression model, our goal is to explore the correlation between environmental factors and the content of flavonoids to provide a fundamental model for soil management [21] in ancient tea gardens, secondary metabolism in ancient tea trees [22], and quality prediction [23] and control of ancient tea [24]. Given the issue that the nomogram model is unable to exhibit multiple outcomes concurrently [24,25], this study successfully enhances the developed visualization system to achieve diverse representations of charts and data. In order to address the problem of inaccurate expression of content range prediction, this study draws on the construction idea of survival curves in the medical field to achieve the construction and display of content prediction curves. This research utilizes the concept of survival curves in the healthcare sector to accomplish the development and presentation of content prognostication curves. This research utilizes a machine–learning model and statistical analysis to establish a system that can predict variations in flavonoids content of ancient tea trees. By training machine learning models, this system acts as a scientific theoretical basis for further investigation into the impact of nonbiological stress conditions on the quality of sun–dried green tea from ancient tea trees. The establishment of this model aims to provide a theoretical basis for the management and quality prediction of ancient tea gardens, as well as quality intervention, and to provide data support for the development of intelligent tea gardens in Yunnan.

## 2. Materials and Methods

### 2.1. Tea and Soil Sample Collection

This study selected eight main cultivation areas of ancient tea trees in Shahe Village, Hetou Village, and Tangfang Village of Laowu Mountain, Zhenyuan County, Pu'er City (N 23° E 100°) as sampling points (Figure 1 presents the specific distribution of eight sampling points). Samples were taken of the selected ancient tea trees for the production of sun–dried green tea and soil. Before excavating the soil of the roots of ancient tea trees, the surface litter and a 4–5 cm layer of soil were removed. The five–point sampling method was used to collect samples from the soil within a vertical depth range of 20 cm and 40 cm. Samples were collected in triplicate at each sampling point, with each sample weighing no less than 200 g. Uniform depth and weight were ensured at each sampling point. A total of

180 samples of ancient tree rhizosphere soil and 90 samples of ancient tree sun–dried green tea were collected for this study. Each sample was numbered and sampling records and sample labels were filled out for each sample.

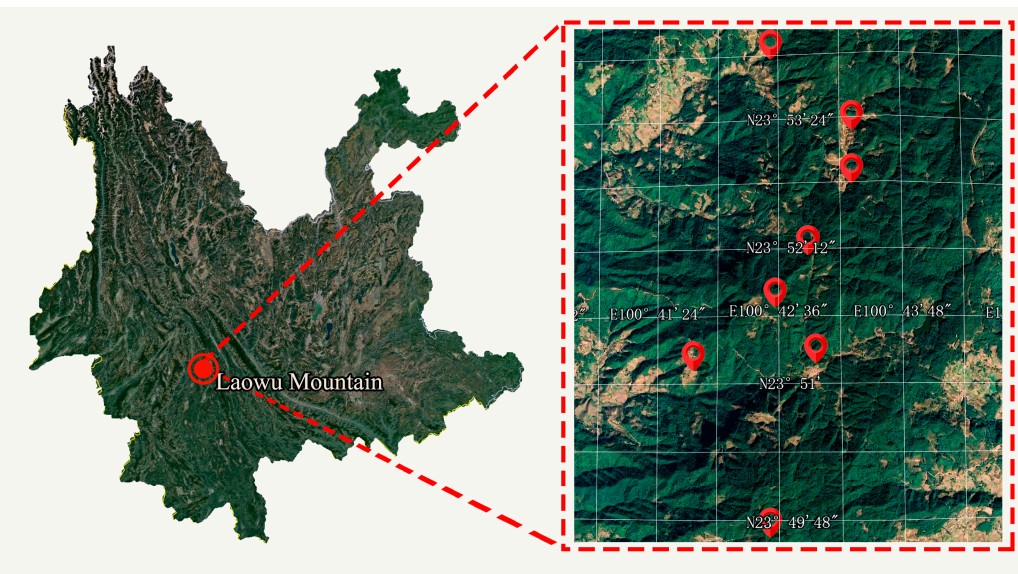

**Figure 1.** Sampling point distribution map.

## 2.2. Detection Methods

The detection method for total flavonoids in tea samples adopted the aluminum chloride colorimetric method. The determination of soil pH was carried out using the potentiometry. The determination of organic matter was obtained from the oxidation of potassium dichromate oxidation spectrophotometric determination of organic carbon multiplied by the constant 1.724. The determination of soil specific gravity was conducted using the pycnometer method, the determination of total phosphorus by alkali fusion−Mo−Sb anti−spectrophotometric method, and the determination of available phosphorus by sodium hydrogen carbonate solution−Mo−Sb anti−spectrophotometric method. The determination of total potassium content was conducted using the ICP−AES method, determination of available potassium content in soil using neutral ferric acetate solution leaching and flame photometry, the determination of total nitrogen by Kjeldahl method, the determination of ammonium nitrogen content using a universal extract−colorimetric method, the determination of nitrate nitrogen content using a phenol−two−sulfonic acid method, and the determination of hydrolytic nitrogen content involved hydrolyzing with a sodium hydroxide solution, adding a boric acid solution, and then using a standard acid titration method for determination.

## 2.3. Statistical Analysis

This study utilized R language version 4.1.2 and used Least Absolute Shrinkage and Selection Operator (LASSO) regression to select modeling factors from 90 samples of ancient tree sun−dried green tea and their corresponding rhizosphere soil samples of ancient tea trees at 20 cm and 40 cm; the flavonoids content was also divided into two labels, >6‰ and >9‰, and three intervals, <6‰, 6‰−9‰, and >9‰, during the modeling process [26,27]. The main components of LASSO regression include reducing the number of variables, creating a penalty function, reducing the coefficients of variables, and forcing some regression coefficients to become zero. LASSO regression is a biased partial method for handling data with multicollinearity, which also has the advantage of subset shrinkage. In addition, LASSO regression is effective in reducing model complexity and the number of required dependent variable types. By controlling parameters, it prevents overfitting [28,29]. In the feature selection phase of this study's LASSO regression model, strongly correlated vari-

ables were selected as modeling factors for nomogram model construction from age of tree, altitude, ammonium nitrogen−20, ammonium nitrogen−40, available phosphorus−20, available phosphorus−40, exchangeable potassium−20, exchangeable potassium−40, hydrolytic nitrogen−20, hydrolytic nitrogen−40, nitrate nitrogen−20, nitrate nitrogen−40, organic carbon−20, organic carbon−40, organic matter−20, organic matter−40, pH−20, pH−40, specificgravity−20, specificgravity−40, total nitrogen−20, total nitrogen−40, total phosphorus−20, total phosphorus−40, total potassium−20, total potassium−40, and tree height, which based on the analysis of the coefficient distribution and cross−validation results of LASSO. In order to confirm the correlation between modeling factors and flavonoid content, this research randomly split the dataset into a training set and a validation set, with a ratio of 8:2, then performed a single factor analysis by employing Cox regression on all variables [30,31]. Additionally, Cox regression also performed a multifactor analysis of the selected modeling factors. *AIC*, which measures the adequacy of statistical models based on entropy, serves as a metric for balancing the complexity of estimated models and the fit to the data. Typically, the model with the lowest *AIC* value is preferred when selecting the best model. So, we also validated the modeling factors using the Akaike Information Criterion (*AIC*) [32,33].

This study utilized ROC curves and calibration curves to evaluate the accuracy and stability of the model. The ROC curve measures the performance of a classification model by calculating the area under the curve, while the calibration curve is used to compare the consistency between actual and predicted results [34,35].

In order to address the limitation of a small modeling dataset, this research additionally introduced the bootstrap method for dataset augmentation [36,37]. The fundamental concept of bootstrap was to obtain numerous samples from the initial dataset, conduct repeated experiments, and create multiple different datasets and then use the empirical distribution of these datasets to substitute the population distribution. This is accomplished by using the random put−back sampling method, where a certain number of samples are taken from the original dataset and, based on these samples, the estimated statistic is calculated and the variance and distribution are estimated based on the results of the calculations.

By introducing the bootstrap method, this research can leverage a larger dataset for modeling, thereby improving the evaluation of accuracy and stability of the model. This can increase model performance and the consistency of the results.

## 3. Results

### 3.1. Factor Selection for Constructing the Model

This study adopted LASSO regression with the aim of selecting the desired number and type of variables before constructing the prediction model. This operation selectively incorporates variables into the machine−learning predictive model, reducing the risk of overfitting for better performance parameters. LASSO regression uses a loss function as follows:

$$J(\theta) = \frac{1}{2m} \sum_{i=1}^{m} \left( h_\theta \left( x^{(i)} \right) - y^{(i)} \right)^2 + \lambda \sum_{j=1}^{n} |\theta_j|, \tag{1}$$

Among them, $\lambda$ is the regularization parameter that primarily controls the complexity of the LASSO model. When $\lambda$ increases, the number of selected variables by the model will decrease, thereby reducing the probability of overfitting to some extent.

In the coefficient distribution plot of LASSO regression, as the penalty coefficient increases, the variable coefficient is gradually reduced due to the penalty term until it reaches 0. The cross−validation plot of LASSO in Figure 2 shows two dashed lines. The first dashed line on the left indicates the minimum mean squared error on the *y*−axis, while the second dashed line on the right represents the standard error level one unit above the minimum mean squared error. With the increase in model complexity, the likelihood of overfitting also increases. Therefore, this study selected the $\lambda$ value when one standard error was used. To address the issue of a small sample size in the dataset, bootstrap resampling was used to expand the dataset during cross−validation. Based on

the coefficient distribution and cross−validation of LASSO regression, this study ultimately selected six factors, namely pH−20, total potassium−20, nitrate nitrogen−20, available phosphorus−40, hydrolytic nitrogen−40, and ammonium nitrogen−40 from 27 variables mentioned above, to establish the prediction model. The research results are consistent with those of Kailing et al., who studied the effects of soil conditions on the flavonoid content of tea plants, as well as with those of Jiling et al., who examined the effects of soil conditions on the flavonoid content of roses [38,39].

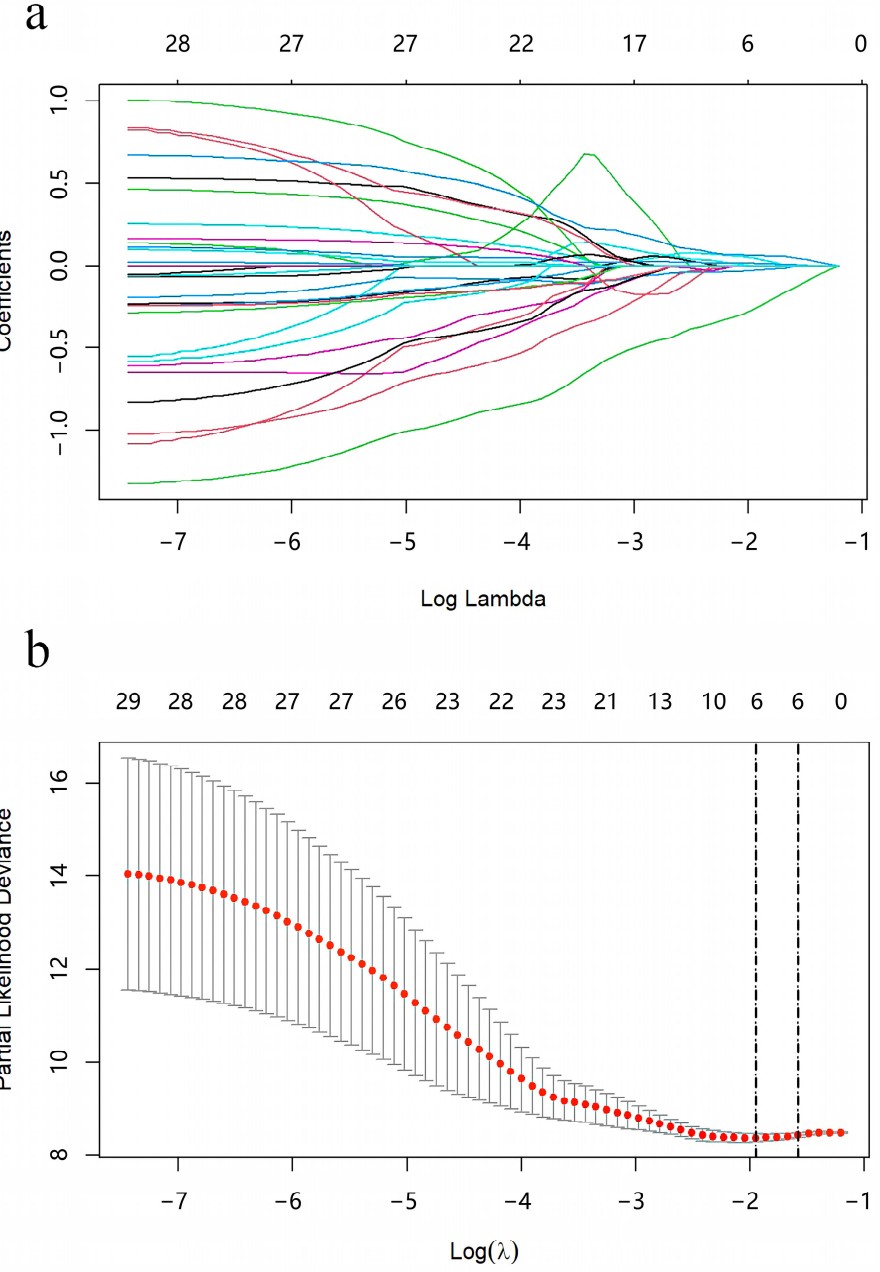

**Figure 2.** Factor screening based on LASSO regression: (**a**) displays the distribution of coefficients in LASSO regression, as the penalty coefficient progressively increases, the colorful lines representing variable coefficient is gradually compressed until it ultimately reaches zero, and (**b**) presents the cross−validation plot, the first dashed line on the left represents the minimum mean squared error on the vertical axis, the second dashed line on the right represents the standard error, which is twice the value of the minimum mean squared error.

### 3.2. Factor Analysis and Model Construction

Cox regression is commonly utilized in clinical medicine for survival analysis and, in this study, Equation (2) illustrates how it was used to assess the collective influence of various factors on fluctuations in flavonoids content.

$$h(t) = h_0(t) \exp(\beta_1 x_1 + \cdots \beta_j x_j), \tag{2}$$

Among them, $h(t)$ is the risk function of the research object, which changes with the change in flavone content; $h_0(t)$ is the intercept of the regression equation; $x_1, x_2, \ldots x_j$ represent independent variables and $\beta_1, \beta_2, \ldots \beta_j$ represent regression coefficients.

To lessen the impact of the dataset on the performance of the model, this research randomly split the dataset into a training set and a validation set, with a ratio of 8:2. Subsequently, Cox regression was employed to examine the correlation between flavonoids content and 27 other variables, conducting both single−factor and multi−factor comparative analysis. The findings of this analysis can be observed in Table 1. It was empirically demonstrated that the LASSO regression identified pH−20, total potassium−20, nitrate nitrogen−20, available phosphorus−40, hydrolytic nitrogen−40, and ammonium nitrogen−40 as significantly impacting the variation in flavonoids content ($p < 0.1$). Moreover, all of these factors exhibited a strong correlation with the fluctuation of flavonoids content.

**Table 1.** Single−factor analysis results and multiple−factor analysis results of flavonoids content changes.

| Characteristics | Single−Factor Analysis Results | | | Multiple−Factor Analysis Results | | |
|---|---|---|---|---|---|---|
| | HR | CI | $p$ | HR | CI | $p$ |
| Age of tree | 1.11 | 0.52–1.56 | 0.71 | | | |
| Altitude | 1.12 | 0.64–1.24 | 0.497 | | | |
| Ammonium nitrogen−20 | 0.97 | 0.84–1.26 | 0.762 | | | |
| Ammonium nitrogen−40 | 0.84 | 1.03–1.39 | 0.023 | 0.52 | 0.36–0.74 | 0 |
| Available phosphorus−20 | 0.99 | 0.98–1.04 | 0.608 | | | |
| Available phosphorus−40 | 0.99 | 1–1.03 | 0.097 | 1.26 | 1.07–1.48 | 0.005 |
| Exchangeable potassium−20 | 1 | 0.77–1.29 | 0.992 | | | |
| Exchangeable potassium−40 | 0.98 | 0.92–1.13 | 0.693 | | | |
| Hydrolytic nitrogen−20 | 1.07 | 0.73–1.2 | 0.595 | | | |
| Hydrolytic nitrogen−40 | 1.42 | 0.53–0.93 | 0.014 | 1.04 | 1.02–1.06 | 0 |
| Nitrate nitrogen−20 | 0.97 | 1.01–1.05 | 0.004 | 0.52 | 0.36–0.74 | 0 |
| Nitrate nitrogen−40 | 1.01 | 0.93–1.04 | 0.633 | | | |
| Organic carbon−20 | 0.9 | 0.85–1.46 | 0.439 | | | |
| Organic carbon−40 | 1 | 0.66–1.53 | 0.982 | | | |
| Organic matter−20 | 1.09 | 0.77–1.1 | 0.341 | | | |
| Organic matter−40 | 0.96 | 0.87–1.24 | 0.656 | | | |
| pH−20 | 1.27 | 0.61–1.02 | 0.07 | 1.26 | 1.07–1.48 | 0.005 |
| pH−40 | 0.97 | 0.71–1.48 | 0.89 | | | |
| Specificgravity−20 | 0.88 | 0.53–2.43 | 0.742 | | | |
| Specificgravity−40 | 0.92 | 0.82–1.44 | 0.55 | | | |
| Total nitrogen−20 | 1.01 | 0.86–1.16 | 0.945 | | | |
| Total nitrogen−40 | 1.04 | 0.85–1.09 | 0.554 | | | |
| Total phosphorus−20 | 0.98 | 0.85–1.22 | 0.838 | | | |
| Total phosphorus−40 | 0.95 | 0.93–1.2 | 0.437 | | | |
| Total potassium−20 | 0.85 | 1–1.38 | 0.047 | 1.04 | 1.02–1.06 | 0 |
| Total potassium−40 | 0.91 | 0.95–1.26 | 0.225 | | | |
| Tree height | 1.19 | 0.69–1.03 | 0.1 | | | |

HR: hazard ratio, CI: confidence interval. −20 represents the measured value of soil quality components in the 20 cm range, −40 represents the measured value of soil quality components in the 40 cm range.

In order to further validate the selected modeling factors for constructing the model, this study included all strongly correlated factors in the modeling process and assessed their Akaike Information Criterion (*AIC*) values. The *AIC* values of these factors were then

compared and analyzed against the *AIC* values associated with the selected modeling factors.

$$AIC = 2k - \ln(L), \tag{3}$$

Among them, $k$ is the model parameter and $L$ is the likelihood function. There are significant differences between the models; the main manifestation is primarily observed in the terms of the likelihood function. When the likelihood functions do not show any notable variations, the complexity of the model comes into play. As the level of model complexity augments (manifested by an escalation in $k$), the likelihood function $L$ exhibits a corresponding proliferation, consequently resulting in a diminutive *AIC*. However, if $k$ becomes excessively large, the growth rate of the likelihood function slows down, causing an increase in the *AIC*. Therefore, an overly complex model tends to result in overfitting. The *AIC* not only enhances the model's fit but also introduces a penalty to minimize the model parameters, thereby reducing the risk of overfitting. Based on the validated results, it is revealed that the *AIC* value of the modeling factors selected through LASSO regression amounts to 463.12, which is a 32.534% decrease compared to the *AIC* value obtained from modeling with all significant factors ($p < 0.1$).

Using two machine−learning models, namely LASSO regression and Cox regression, this study conducted a comprehensive screening of factors that exhibited a robust correlation with the flavonoid content found in sun−dried green tea derived from ancient trees. Afterwards, the nomogram model was employed to effectively visualize and articulate the intricate interplay among these factors. Nomogram model is based on multiple−factor regression analysis, integrating multiple modeling factors. By plotting lines with scale lines on the same plane, the model visually demonstrated the relationships between variables. Each variable factor in the modeling was assigned a corresponding score based on its importance in Figure 3. The anticipated likelihood of the outcome variable was derived by employing the functional transformation correlation between the aggregate score and the outcome variable. Specifically, in order to predict using the machine−learning model, a composite score was calculated by summing the values of pH−20, total potassium−20, nitrate nitrogen−20, available phosphorus−40, hydrolytic nitrogen−40, and ammonium nitrogen−40 on the parameter "Points". Finally, the predicted outcome scores were obtained by looking up the corresponding values on the "Total Points". From the figure, it is evident that the content of hydrolytic nitrogen−40 in the soil had the most pronounced impact on the flavonoids, followed by nitrate nitrogen−20, total potassium−20, available phosphorus−40, and ammonium nitrogen−40, respectively. Amongst these factors, pH−20 had the least influence on the flavonoid content. The units for all factors, except pH, are measured in mg/kg. Furthermore, the graphical representation unveiled positive correlation between flavonoids content and pH−20, as well as hydrolytic nitrogen−40. Conversely, a negative correlation was observed between flavonoids and total potassium−20, nitrate nitrogen−20, available phosphorus−40, and ammonium nitrogen−40. Notably, these findings align with the findings of Rozy's investigation, highlighting a negative relationship between ammonium nitrogen and the overall accumulation of flavonoids in Amaranthus plants [40].

### 3.3. Assessing the Stability of the Model

The calibration curve serves as a fundamental instrument deployed for assessing the extent of congruity between the projected outcomes of the model and the prevailing real−world circumstances [41,42]. In this study, it was used to evaluate the fitting degree between the changes in flavonoids content by the nomogram under abiotic stress. The $x-$axis serves as an appraised portrayal of the envisaged levels of flavonoids, while the $y-$axis portrays the authentic flavonoids content. The diagonal dotted line represents a perfect prediction by an ideal model. The orange solid line represents the performance of the nomogram model, of which a closer fit to the diagonal dotted line represents a perfect prediction by an ideal model. To ensure the accuracy of this research, bootstrap was used to resample the sample, and the resampling times were 3000 times. From the calibration

curve of the nomogram model in this study, it was evident that the actual calibration curve of the prediction model closely matched the ideal calibration curve, and there was obvious consistency in the training set and verification set. The model corresponded to the verification set Figures 4c and 4d in the training set Figures 4a and 4b, respectively.

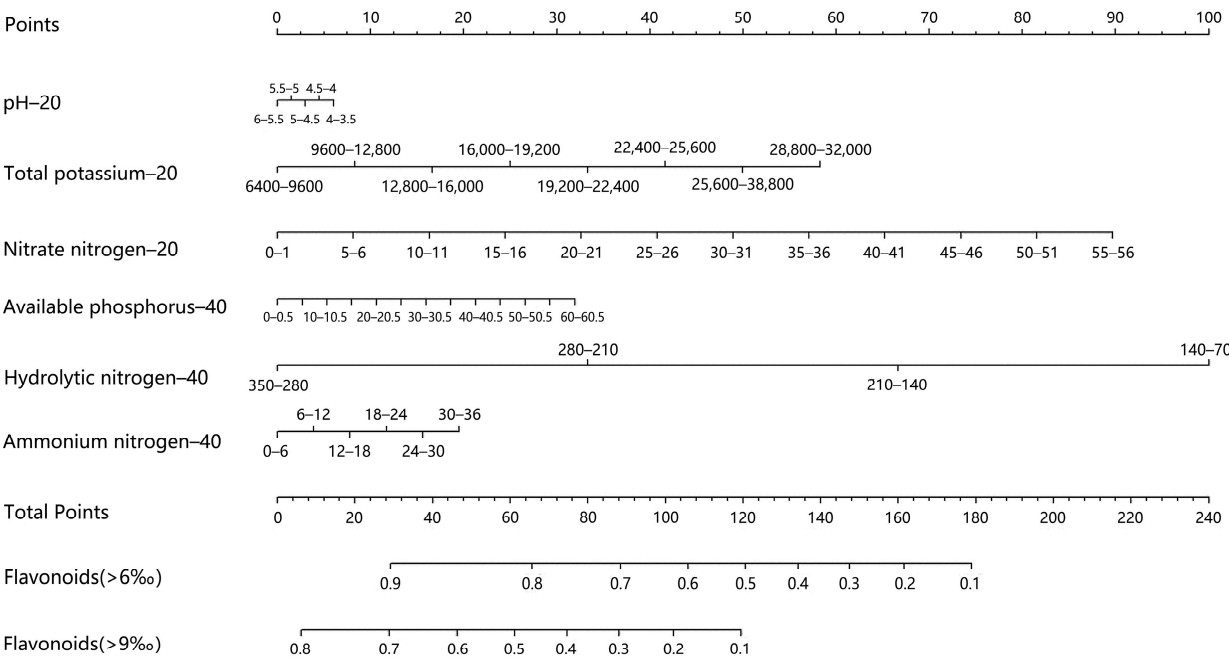

**Figure 3.** Nomogram model predicts the relationship between the range of variation in six strongly correlated factors and their corresponding flavonoids content.

### 3.4. Model Accuracy Assessment

The ROC curve is a metric used to evaluate the performance of a model at different thresholds, initially used for radar signal analysis and now mainly applied to measure the performance of machine learning models. In the ROC curve, the closer the position of the curve is to the upper left, the better its prediction. For model accuracy assessment, we classified the real situation into two categories, positive and negative. Based on the prediction results, we further classified the samples into four categories, i.e., true positive (*TP*), false negative (*FN*), false positive (*FP*), and true negative (*TN*) [43]. When the predicted value is exactly the same as the true value, it is called true positive or true negative and, when there is a difference between the predicted value and the true value, it is called false negative or false positive. In the ROC curve, the vertical co−ordinate denotes the true positive rate (*TPR*), where a larger value proves a more accurate prediction, and the horizontal co−ordinate denotes the false positive rate (*FPR*), where a smaller value proves a more accurate prediction.

$$TPR = \frac{TP}{TP + FN} \tag{4}$$

$$FPR = \frac{FP}{FP + TN} \tag{5}$$

Since the ROC curve cannot quantitatively evaluate the model performance, the model performance is evaluated by calculating the area under curve (AUC) [44]; the area under the ROC curve reflects the classification ability, with a larger value indicating a stronger classification ability. It is generally considered that, when the AUC value is ≥0.7, it indicates that the model has good differentiation ability. When the flavonoids content is above 6‰ in this model, the AUC values achieved for the training set and validation set are 0.8121 and 0.792, respectively. However, if the flavonoids content surpasses 9‰, the AUC values for the training set and validation set improve to 0.877 and 0.889, respectively. These results

indicate that the model's prediction performance, as shown in in Figure 5, is promising, as the AUC values for both sets are above 0.7 (Figure 5a shows the training set and Figure 5b shows the validation set).

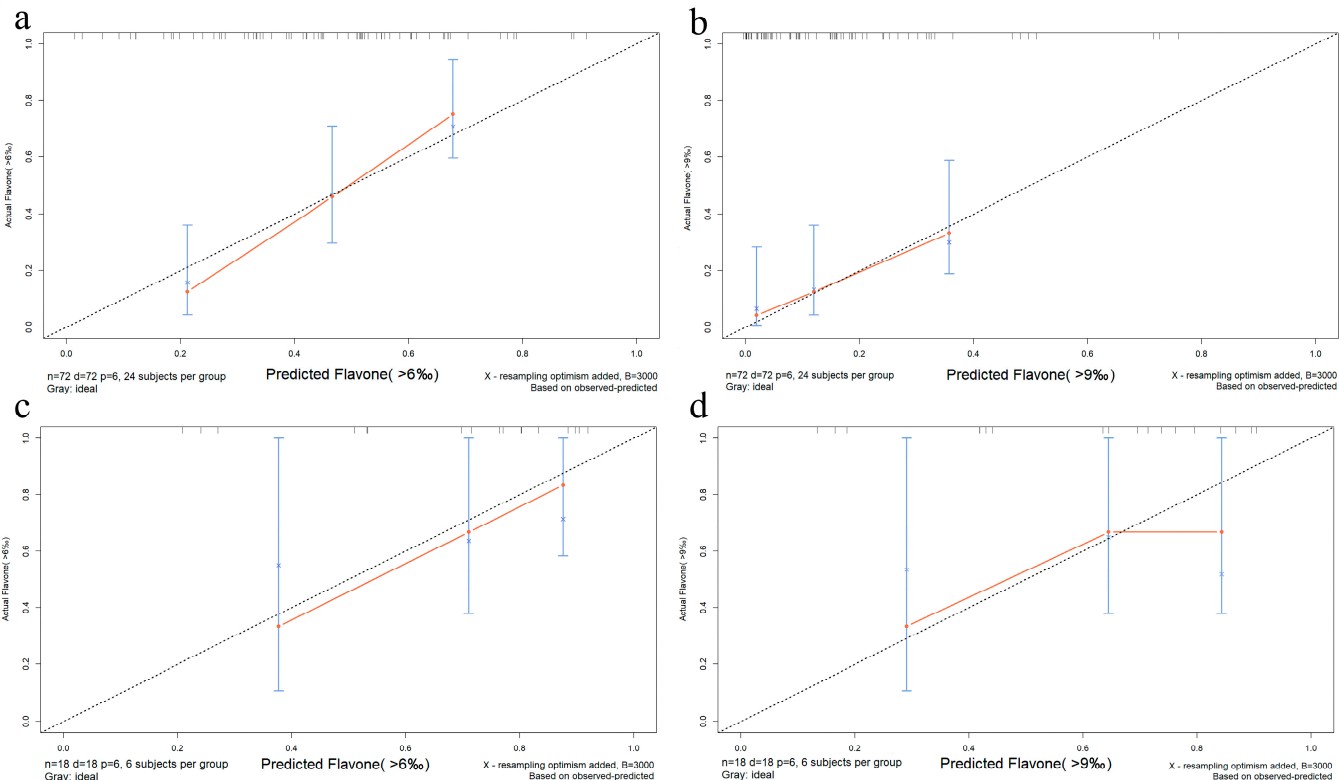

**Figure 4.** Calibration curve. (**a**) shows the calibration curve of the training set when the flavonoids content is >6‰, (**b**) shows the calibration curve of the training set when the flavonoids content is >9‰, (**c**) shows the calibration curve of the validation set when the flavonoids content is >6‰, and (**d**) shows the calibration curve of the validation set when the flavonoids content is >9‰. In these figures, the diagonal dotted line represents a perfect prediction by an ideal model, and the orange solid line represents the performance of the nomogram model, of which a closer fit to the diagonal dotted line represents a perfect prediction by an ideal model.

### 3.5. Construction of a System and Testing of a Model

The efficiency of the flavonoids content calculation in tea solely based on machine−learning models remains an issue in practical implementations; in order to further solve this problem, this study developed a corresponding visualization system based on machine−learning models. The system aims to enable faster and more accurate prediction of flavonoids content in tea. The developed system in Figure 6 consists of five main modules: information input, survival plot, predicted survival, numerical summary, and model summary.

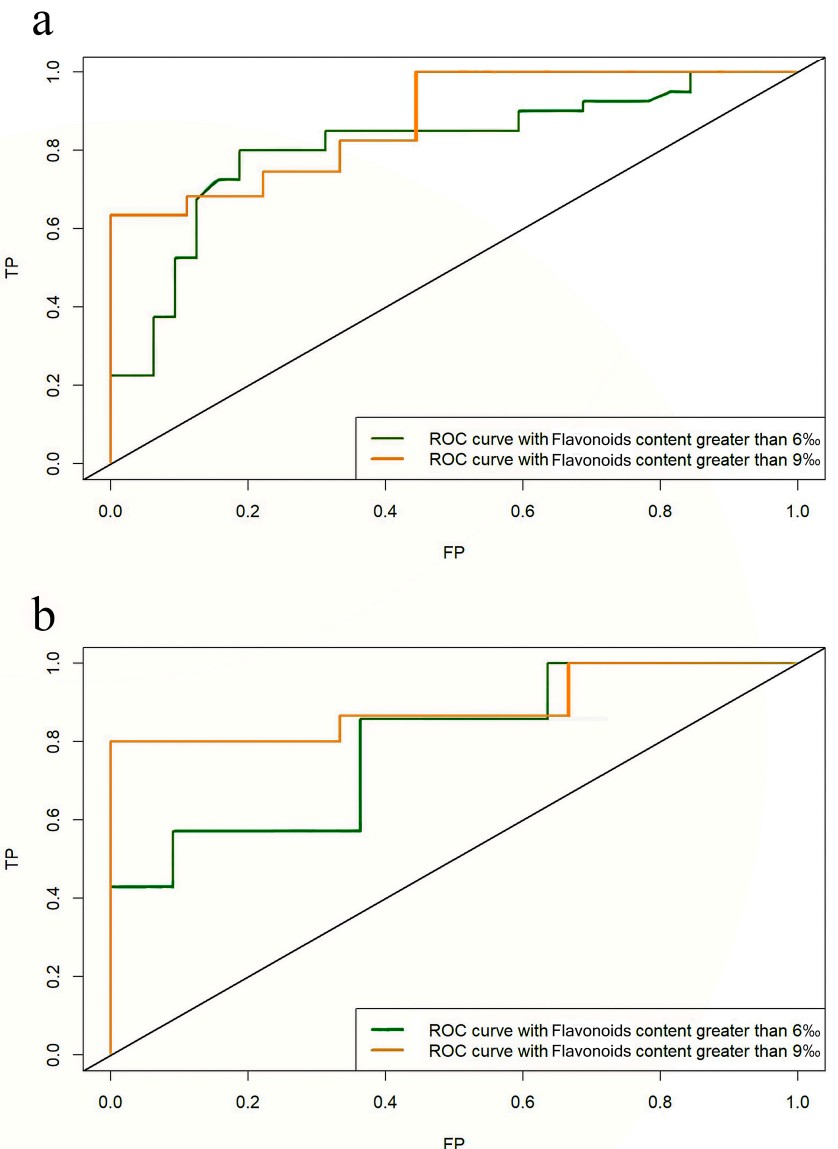

**Figure 5.** ROC curve analysis. (**a**) shows the training set and (**b**) shows the validation set. When the flavonoids content is above 6‰ in this model, the AUC values achieved for the training set and validation set are 0.8121 and 0.792, respectively. However, if the flavonoids content surpasses 9‰, the AUC values for the training set and validation set improve to 0.877 and 0.889, respectively. The horizontal axis FP represents false positive and the vertical axis TP represents true positive, and the black line is the baseline (minimum standard). The farther the ROC curve is from the baseline, the better the predictive performance of the model.

The information input module is utilized to enter the values of pH−20, total potassium −20, nitrate nitrogen−20, available phosphorus−40, hydrolytic nitrogen−40, and ammonium nitrogen−40. The values for the six factors are entered and predicted using the "Predict" button.

The survival plot module in Figure 6a is a graphical module for displaying the predicted probability of producing differences in flavonoids content under different abiotic stress environmental conditions. The module visualizes the predicted range of flavonoids content using a curve format; the range of predicted flavonoid content is depicted by various colored curves, with each curve representing a different environmental condition. The content prediction curves generated by the survival plot module will vary when the predicted range of flavonoids content remains consistent across different abiotic stresses. The color of the curve becomes lighter as the probability of prediction decreases. Through

the comparison of these content prediction curves, a more precise understanding of the soil condition in ancient tea plantations can be attained, enabling the implementation of appropriate management strategies.

In Figure 6b, the survival module's predicted values are visibly depicted as co−ordinates on the graph. When the mouse cursor is positioned over a particular point, the corresponding values for pH−20, total potassium−20, nitrate nitrogen−20, available phosphorus−40, hydrolytic nitrogen−40, ammonium nitrogen−40, prediction, lower bound, and upper bound are all displayed on the chart. Up to 11 predictions can be shown in the table at the same time, and predictions exceeding this number will overwrite the initial prediction.

In the numerical summary module in Figure 6c, the detailed data of all forecasts will be clearly displayed, and the forecast results covered in the survival plot will also be displayed in this module. The core component of this system is the model summary module depicted in Figure 6d, which encompasses all the parameters essential for constructing the machine–learning model employed in this instance.

In the process of using the system, users can input the values of pH−20, total potassium−20, nitrate nitrogen−20, available phosphorus−40, hydrolytic nitrogen−40, and ammonium nitrogen−40 in the information input module and click the Predict button to obtain the prediction result. The prediction results include prediction, lower bound, and upper bound. According to the original machine−learning model, when the forecast achieves a specific threshold of more than 0.5, the flavonoids content is considered to be more than 6‰ of the specific value. When the prediction reaches a specific threshold of more than 0.75, the content of total flavonoids is considered to be more than 9% of the specific value.

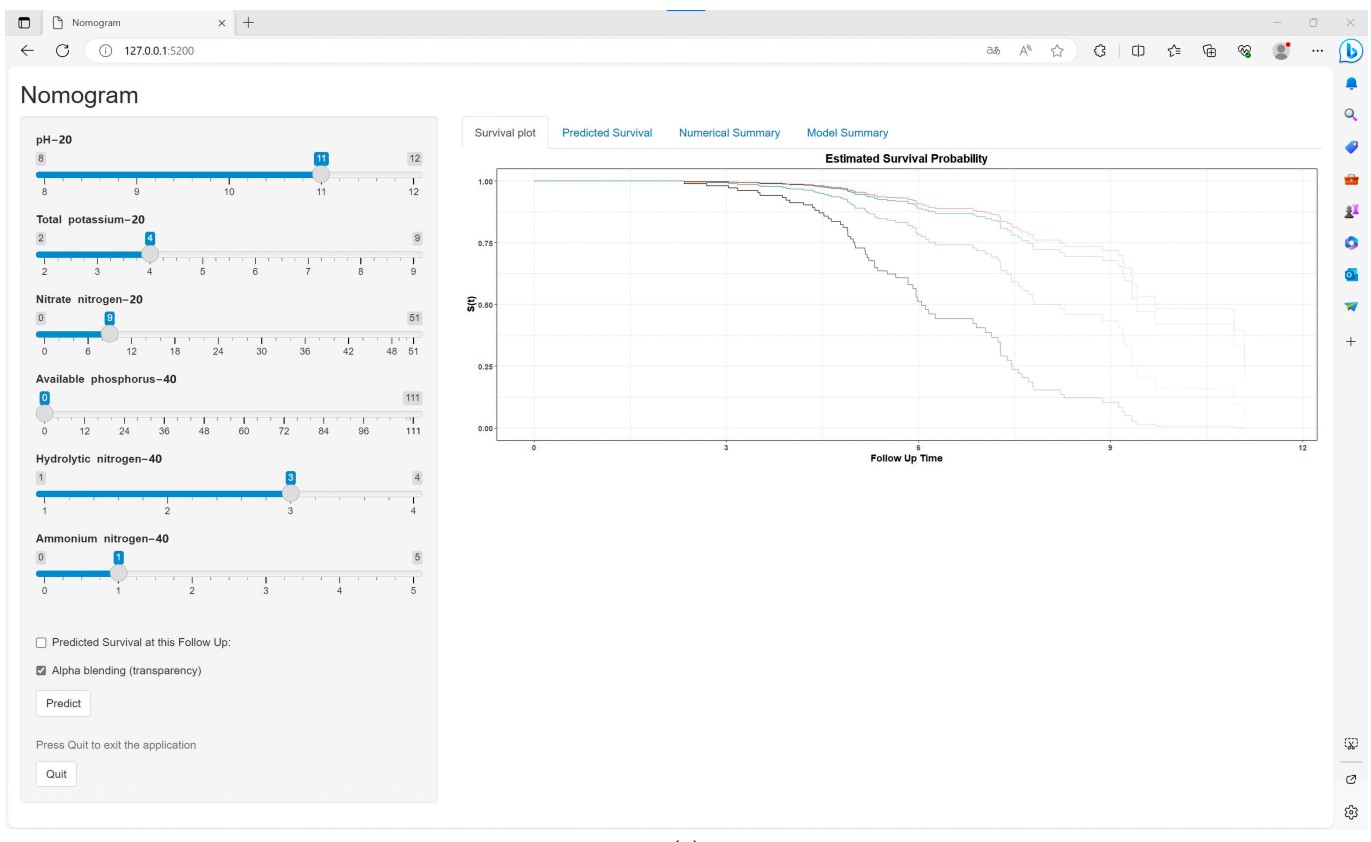

(**a**)

**Figure 6.** *Cont.*

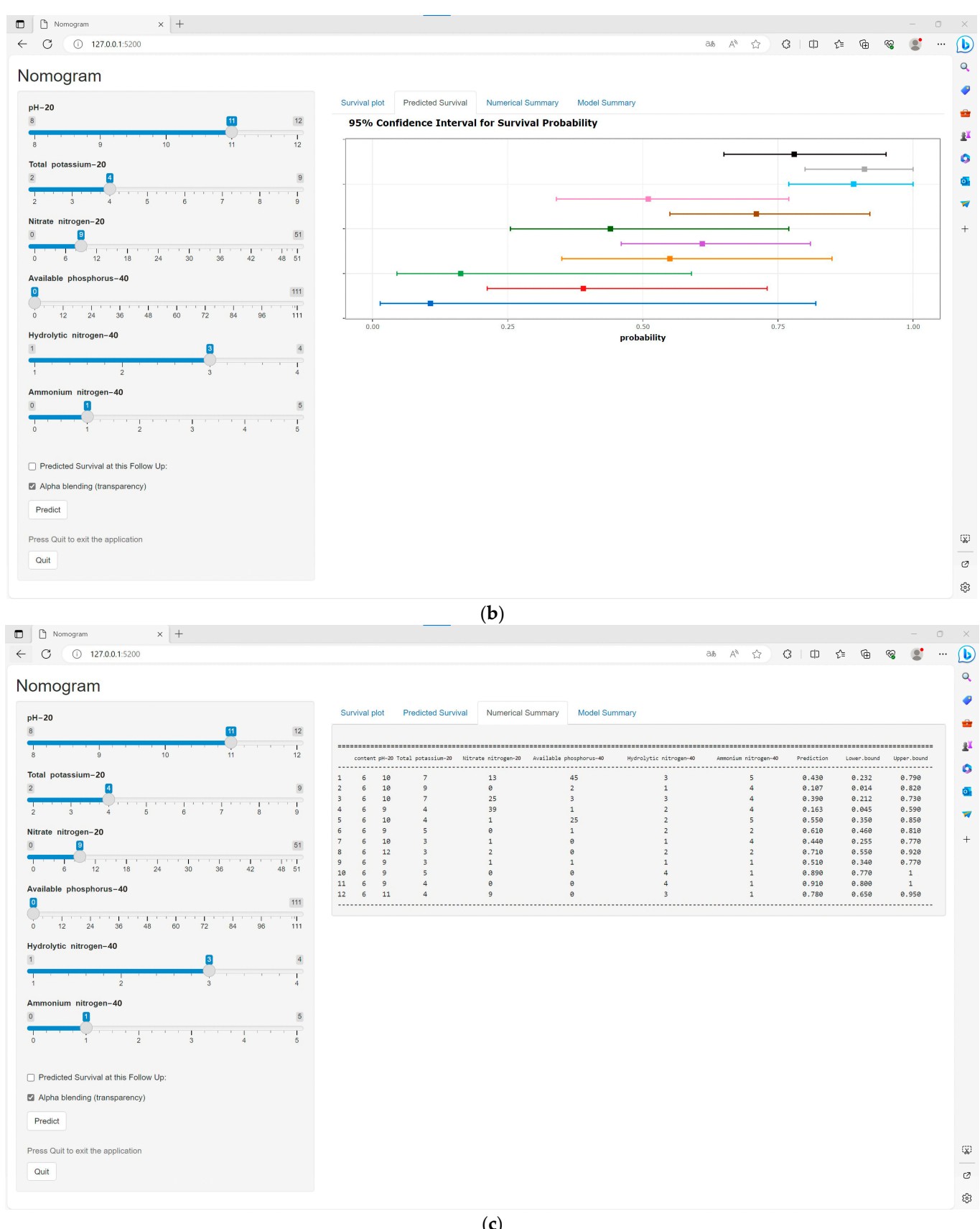

(**b**)

(**c**)

**Figure 6.** *Cont.*

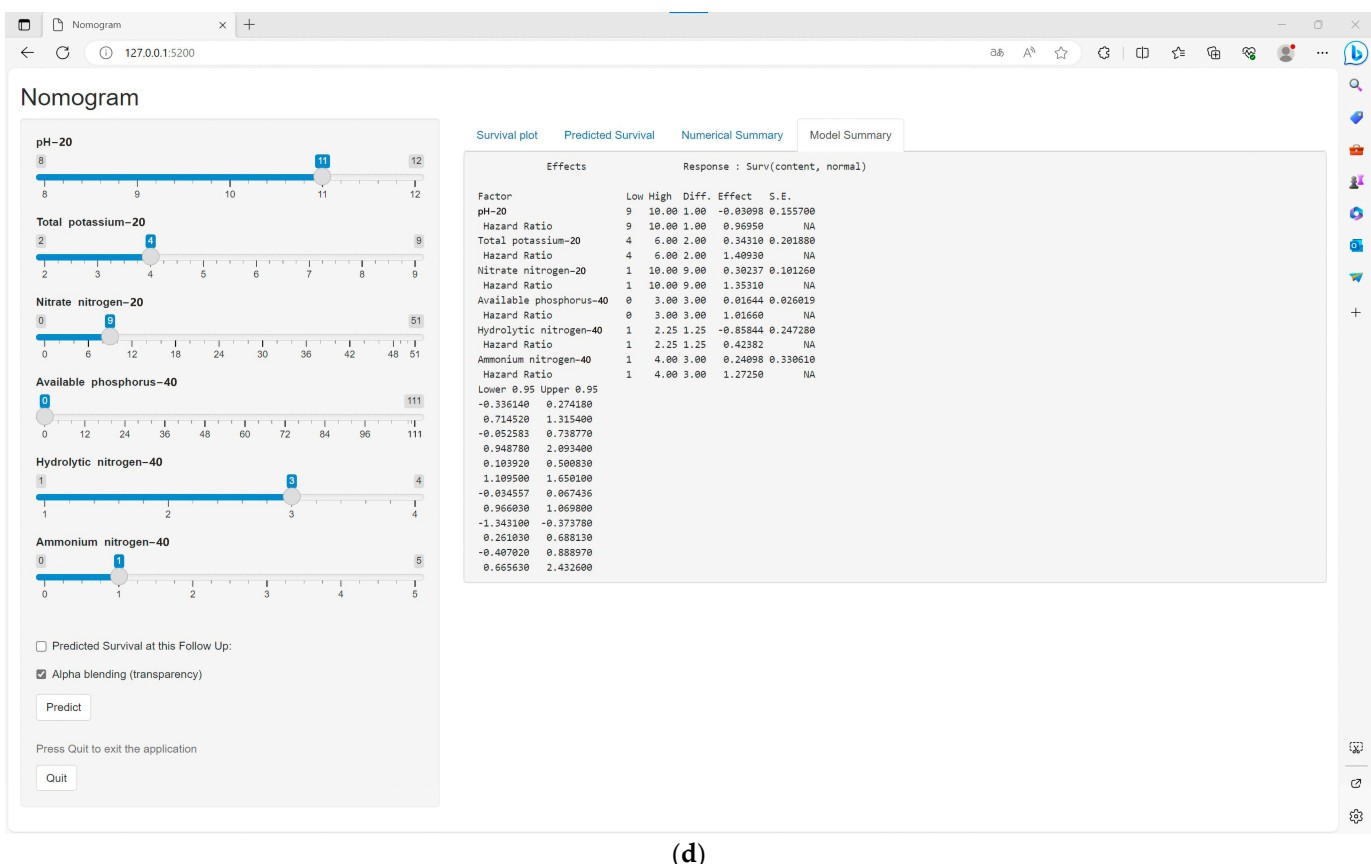

(**d**)

**Figure 6.** Main interface of the visualization system for flavonoids content prediction. (**a**) is survival plot module, (**b**) is predicted survival module, (**c**) is numerical summary module, and (**d**) is model summary module. (**a**) visually presents the predicted range of flavonoids content in a curve form. The different colors of the curves represent the predicted range of flavonoids content under different environmental conditions. As the prediction probability decreases, the color of the curve becomes lighter and lighter. (**b**) visually presents the final predicted values as plotted coordinates. The use of various colored lines differentiates between different prediction outcomes.

In this study, pH, total potassium, nitrate nitrogen in the inter−root soil at a vertical depth of 20 cm, available phosphorus, hydrolytic nitrogen, and ammonium nitrogen at a vertical depth of 40 cm at the same location in the Laowu Mountain Tea Tree of Zhenyuan County, Pu'er City and corresponding total flavonoids in the tea leaves of the Laowu Mountain Tea Tree were selected as the parameters for validation and, overall, 12 sets of data were taken advantage of for the external test of the model. The ultimate outcome showed that 10 groups were correctly predicted and 2 groups of experimental data were incorrectly predicted, with a total accuracy of 83.33% in Table 2. The prediction model for flavonoids content attained a significant level of accuracy, signifying its effectiveness.

**Table 2.** External validation results of the flavonoids prediction model using 12 sets of data.

| Flavonoids (‰) | pH−20 | Total Potassium−20 (mg/kg) | Nitrate Nitrogen−20 (mg/kg) | Available Phosphorus−40 (mg/kg) | Hydrolytic Nitrogen−40 (mg/kg) | Ammonium Nitrogen−40 (mg/kg) | Grade | Correct |
|---|---|---|---|---|---|---|---|---|
| 2.867 | 5.17 | 22,870 | 13.431 | 22.610 | 250.600 | 31.200 | 0.430 | √ |
| 3.175 | 5.67 | 20,018 | 15.438 | 23.500 | 225.700 | 28.400 | 0.107 | √ |
| 4.984 | 5.28 | 24,511 | 25.805 | 2.780 | 229.400 | 29.500 | 0.390 | √ |
| 5.200 | 4.79 | 14,594 | 39.125 | 0.790 | 202.800 | 25.000 | 0.163 | √ |
| 6.856 | 5.19 | 14,600 | 1.430 | 12.900 | 141.000 | 32.100 | 0.550 | √ |
| 7.042 | 4.99 | 19,100 | 0.567 | 0.860 | 177.000 | 14.600 | 0.610 | |
| 7.143 | 5.43 | 11,263 | 1.790 | <0.5 | 88.300 | 27.000 | 0.440 | |
| 8.108 | 6.05 | 12,300 | 2.840 | <0.5 | 181.000 | 13.400 | 0.710 | √ |
| 9.106 | 4.71 | 10,288 | 1.767 | 0.650 | 71.200 | 10.600 | 0.510 | |
| 9.191 | 4.82 | 17,279 | 0.856 | <0.5 | 292.300 | 7.800 | 0.890 | √ |
| 9.319 | 4.79 | 15,095 | 0.903 | <0.5 | 3177.300 | 7.800 | 0.910 | √ |
| 10.927 | 5.77 | 13,883 | 9.709 | <0.5 | 227.700 | 10.400 | 0.780 | √ |

## 4. Discussion

This study used LASSO−Cox regression to select six modeling factors from 27 variables of ancient tree sun−dried green tea and their corresponding rhizosphere soil samples of ancient tea trees at 20 cm and 40 cm; the flavonoids content was also divided into two labels, >6‰ and >9‰, and three intervals, <6‰, 6‰−9‰, and >9‰, during the modeling process. In this study, there is a positive correlation between flavonoids content and pH−20 and hydrolyzed nitrogen−40. There is a negative correlation between flavonoids content and total potassium−20, nitrate nitrogen−20, available phosphorus−40, and ammonium nitrogen−40. Previous studies have shown a negative relationship between ammonium nitrogen and the overall accumulation of flavonoids in Amaranthus plants [40]. *Clinacanthus nutans* with high flavonoids content corresponds to slightly acidic soil pH [45]. The above−mentioned results are consistent with the results of this study. Currently, there is limited research on the interaction between tea flavonoids and soil factors, and this study focuses on soil samples from an ancient tea garden, making it unable to determine the specific impact of individual variables on flavonoids content. Further research is necessary to conduct pot experiments aimed at investigating the individual impact of soil factors on flavonoids content. This could be achieved by carefully manipulating and controlling a single variable at a time, while ensuring that other relevant factors remain constant.

At present, traditional line graph models have problems such as inability to analyze and predict data quickly and to display multiple results simultaneously. In this study, a LASSO−Cox regression model was used to select six significant variables for constructing a column–line chart model. This model can provide data and theoretical support for soil management in the Laowu Mountain ancient tea tree gardens; it can also predict the content of catechins in the ancient tree sun−dried green tea of Laowu Mountain based on soil conditions, while ensuring balanced taste; the flavonoids content can be increased appropriately [46], thereby enhancing the antioxidant [47], immune regulation [48], anti-cancer [49], anti−inflammatory, and detoxifying effects of ancient tea. However, due to the concentrated collection of specimens in the ancient tea gardens of Laowu Mountain during the spring season, there is a lack of sample data from other seasons and regions. Yunnan has a vast and diverse tea−growing region, with each tea−growing area having its own microclimate and soil conditions. Therefore, it is crucial to supplement the data on soil conditions and flavonoid content from other production regions. Future work is needed for collecting data on different seasons of Laowu Mountain and other sun−dried green−tea−producing regions in order to improve the data system and establish a more universal predictive model. Tea contains a variety of substances and they collectively determine the taste and effectiveness of tea. In the future, our team will also make predictions on other major components in order to achieve a comprehensive evaluation of its drinking, medical, and industrial value. Through the establishment of various intelligent models, we aim to provide the necessary data foundation for the smartification of tea gardens and IoT device systems. This will enable the early perception and prediction of soil conditions and related tea quality, allowing for early intervention to ensure the quality of tea leaves.

## 5. Conclusions

The study chose different altitudes, tree ages, tree heights, and soil compositions in the Laowu Mountain, Zhenyuan County, Pu'er City Yunnan Province to analyze the flavonoids content in ancient tree sun−dried green tea. The LASSO−Cox regression model selected six elements, namely pH−20, total potassium−20, nitrate nitrogen−20, available phosphorus−40, hydrolytic nitrogen−40, and ammonium nitrogen−40, to establish a predictive model for the flavonoids content in ancient tree sun−dried raw green tea in this region. In this model, both the training and validation sets' values surpass 0.6, demonstrating high reliability. This study innovatively offers a tool to preliminarily predict the content of total flavonoids in tea leaves through effective soil components related to tea tree. The validation results from all aspects demonstrate its good predictive ability. Compared to the construction of a conventional nomogram model, this study has produced a flavonoids content prediction system corresponding to the nomogram model, which addresses the inability of the column–line chart model to perform analysis and prediction quickly. This prediction system not only allows for a rapid prediction of flavonoids content but also presents the predicted values in the system in both co−ordinates and numerical form. In the survival plot, the estimated values for each sample are indicated with different colors, allowing experimenters to compare and analyze the estimated values of flavonoids content for several samples. Next, the system users can make preliminary predictions on the contents of other components in tea, such as soluble sugars and free amino acids, based on factors like soil, altitude, and age of the tea plants. The quality of the teas comprehensively appraised can be utilized by these predictions. The quality of tea leaves can also be regulated by adjusting the management measures of tea gardens, thereby changing the proportions of the components in tea leaves to meet the production demand.

**Author Contributions:** Conceptualization, visualization, writing—original draft preparation, L.L. and Y.W.; data curation, methodology, software, H.W., J.H. and Q.W.; validation, formal analysis, J.X., Y.X. and W.Y.; investigation, resources, S.C. and L.T.; conceptualization, writing—review and editing, funding acquisition, B.W. and X.W. All authors have read and agreed to the published version of the manuscript.

**Funding:** This research was funded by Research on the formation mechanism of the characteristic quality of tea in the ecological driven Laowu Mountain small tea producing area (founder: Yunnan Provincial Science and Technology Department, founding number: 202301BD070001–070); Yunnan Provincial Development and Reform Commission's "Ten Thousand Talents Program" for Industrial and Technological Leading Personnel (founder: People's Government of Yunnan Province, founding number: YNWR–CYJS–2018–009); Integration and Demonstration of Key Technologies for Improving Quality and Efficiency of the Tea Industry in Lvchun County under the National Key R&D Project (founder: Key Technologies Research and Development Program, founding number: 2022YFD1601803); Development and demonstration of intelligent agricultural data sensing technology and equipment in plateau mountainous areas. (founder: Major Science and Technology Projects in Yunnan Province, founding number: 202302AE09002001); Study on the screening mechanism of phenotypic plasticity characteristics of large–leaf tea plants in Yunnan driven by AI based on data fusion (founder: Yunnan Provincial Science and Technology Department, founding number: 202301AS070083); Yunnan Menghai County Smart Tea Industry Science and Technology Mission (founder: Yunnan Provincial Science and Technology Department, founding number: 202304BI090013); and Preparation and Performance Study of Zijuan tea anthocyanins–Soy Protein Isolate Based Intelligent Composite Packing Film (founder: Yunnan Provincial Department of Education, founding number: 2023Y1028).

**Institutional Review Board Statement:** Not applicable.

**Data Availability Statement:** The original contributions presented in the study are included in the article. Further inquiries can be directed to the corresponding authors.

**Acknowledgments:** We thank the editors and the anonymous reviewers for their valuable comments and suggestions.

**Conflicts of Interest:** The authors declare no conflicts of interest. In addition, author Lin Tao was employed by the company Pu'er Wenbang Tea Co., Ltd., he was only involved in survey sampling in this study and was not involved in providing funding. The remaining authors declare that the research was conducted in the absence of any commercial or financial relationships that could be construed as a potential conflict of interest.

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
