# Peer review of "Prediction Model of Flavonoids Content in Ancient Tree Sun−Dried Green Tea under Abiotic Stress Based on LASSO−Cox"

_agriculture, doi:10.3390/agriculture14020296_

Round 1
Reviewer 1 Report
Comments and Suggestions for Authors
This study focusses on developing a model for predicting flavonoids content in ancient tree sun dried green tea. The manuscript contains good technical merit. However, there are several important comments that the authors need to address before their manuscript can be published in the Agriculture journal.
Title should be Prediction Model of Flavonoids Content in Ancient Tree Sun dried Green Tea under Abiotic Stress Based on LASSO regression
2. Materials and Methods
2.1. Experimental materials
In line 109, Please check the sentence “90 samples of ancient tree tea sun-dried green tea 8were collected for this study.”
In line 112 – 126 Please check using capital letters.
2.2. Statistical analysis
In my opinion, I think that the computer specs do not need to be reported because you are not studying the processing time performance.
How many samples are there for training the model. Did you split the sample into a training and testing set? Please explain.
3. Results
Figure 5 is not clear. Please enhance the quality of this figure.
Author Response
Dear Reviewer,
Thanks very much for taking your time to review this manuscript. I really appreciate all your comments and suggestions! Please find my itemized responses in below and my revisions/corrections in the re-submitted files.
Here are the journal comments from the reviewer:
1.Title should be Prediction Model of Flavonoids Content in Ancient Tree Sun dried Green Tea under Abiotic Stress Based on LASSO regression.
We are thankful for your guidance, due to the utilization of LASSO regression and Cox regression models in our article, per your suggestion, we have modified the title to "Prediction Model of Flavonoids Content in Ancient Tree Sun-dried Green Tea under Abiotic Stress Based on LASSO-Cox".
- In line 109, Please check the sentence “90 samples of ancient tree tea sun-dried green tea 8were collected for this study.”
Thank you for your reminder. We have already changed “90 samples of ancient tree tea sun-dried green tea 8were collected for this study” to "90 samples of ancient tree sun-dried green tea were collected for this study "in line 119-120.
3.In line 112 – 126 Please check using capital letters.
Thank you for your reminder, we have examined the passages originally at 112-126 verbatim. Details of the changes are set out below:
In line 126-127, we changed “Potentiometry” to "potentiometry".
In line 128-129, we changed “Potassium Dichromate Oxidation Spectrophotometric” to "potassium dichromate oxidation spectrophotometric".
In line 131, we changed “Anti” to "anti".
In line 132, we changed “Sodium” to "sodium".
In line 134-135, we changed “Determination” to "determination".
In line 137, we changed “Universal” to "universal".
- In my opinion, I think that the computer specs do not need to be reported because you are not studying the processing time performance.
We really appreciate your advice, and now we have made cuts to the computer’s performance and configuration, only retaining the version of R language used in the study. The specific changes are presented in line 160.
- How many samples are there for training the model. Did you split the sample into a training and testing set? Please explain.
We are thankful for your guidance, a total of 90 ancient sun-dried tea and its corresponding rhizosphere soil samples of ancient tea trees at 20cm and 40cm were used to train the model. And this research randomly split the dataset into a training set and a validation set, with a ratio of 8:2. In addition, we made supplementary explanations on line 160-163 and line 178-180 in the manuscript respectively.
- Figure 5 is not clear. Please enhance the quality of this figure.
Thank you for your reminder. We have improved the clarity of all the figures, including the original figures 5 (now figures 6).
We would like also to thank you for allowing us to resubmit a revised copy of the manuscript.
We hope that the revised manuscript is accepted for publication in the Journal of Agriculture-Basel. Thanks again!
Kind regards,
Baijuan Wang
E-Mail: wangbaijuan2023@163.com

Reviewer 2 Report
Comments and Suggestions for Authors
In my opinion the quality of the paper is generally fine
1/The abstract is fine but lacks in my opinion some numerical data regarding the estimated and measured traits
2/The introduction is somehow long and preferably length should be reduced in some longer sections
3/Materials and methods section is clear
4/Results: the section is fine butthe quality of most of the fugures should be improved for claruty
5/Discussion: the section should be imroved inserting clear comparison
Comments on the Quality of English Language
Generally the quality id good and clear
Author Response
Dear Reviewer,
Thanks very much for taking your time to review this manuscript. I really appreciate all your comments and suggestions! Please find my itemized responses in below and my revisions/corrections in the re-submitted files.
Here are the journal comments from the reviewer:
- The abstract is fine but lacks in my opinion some numerical data regarding the estimated and measured traits.
We are thankful for your guidance. According to your advice, we have added” In this prediction model, when the flavonoid content>6‰, the area under the curve of the training set and validation set were 0.8121 and 0.792, when the flavonoid content>9‰, the area under the curve of the training set and validation set were 0.877 and 0.889, demonstrating good consistency.” in line 28-30, which made our research more convincing.
- The introduction is somehow long and preferably length should be reduced in some longer sections.
Thank you for your reminder. We have removed the following sentences:
“With the increasing investigation on total flavonoids in recent years, it has been discovered that total flavonoids are a hopeful framework for the creation of novel medications against dengue fever. As a core structure, total flavonoids have great potential as novel antiviral agents in future development.” in line 54-58.
"This shows that total flavonoids can be widely not only in industries such as pharmaceuticals, food, and cosmetics as natural antioxidants, natural colors, natural food preservatives, etc [13], but also have great potential for development in the industrial sec-tor. "in line 60-63.
“The nomogram model utilized in this research is an extremely instinctive mathematical model of machine learning that can forecast particular results by amalgamating multiple factors [15]. Compared to other predictive models, machine learning model typically presents its results in a more intuitive form such as graphs and has faster model construction speed and lower running latency. As a trustworthy and us-er-friendly tool commonly utilized in the medical domain, machine learning model can offer more precise forecasting and control techniques for agricultural production, thus advancing the sustainability of agriculture [16,17].” in line 75-82.
At the same time, we found that after removing these sentences, the structure of the article became clearer while the content remained substantial.
- Materials and methods section is clear.
Thank you for your endorsement.
- Results: the section is fine but the quality of most of the fugures should be improved for claruty.
Thank you for your reminder. We have improved the clarity of all the figures.
- Discussion: the section should be imroved inserting clear comparison.
We are thankful for your guidance, we have added refinements to the discussion section in line 471-493. The following is our discussion part:
At present, traditional line graph models have problems such as inability to analyze and predict data quickly, and to display multiple results simultaneously. In this study, a LASSO-Cox regression model was used to select six significant variables for con-structing a column-line chart model. This model can provide data and theoretical sup-port for soil management in the Laowu Mountain ancient tea trees garden, it can also predict the content of catechins in the ancient tree sun-dried green tea of Laowu Mountain based on soil conditions, 3while ensuring balanced taste, the flavonoids content can be increased appropriately [45], thereby enhancing the antioxidant [46], immune regulation [47], anti-cancer [48], anti-inflammatory, detoxifying effects of an-cient tea. However, due to the concentrated collection of specimens in the ancient tea gardens of Laowu Mountain during the spring season, there is a lack of sample data from other seasons and regions. Yunnan has a vast and diverse tea-growing region, with each tea-growing area having its own microclimate and soil conditions. Therefore, it is crucial to supplement the data on soil conditions and flavonoid content from other production regions. The next step for the team is to collect data on different seasons of Laowu Mountain and other sun-dried green tea producing regions, in order to improve the data system and establish a more universal predictive model. Tea contains a variety of substances, and they collectively determine the taste and effectiveness of tea. In the future, our team will also make predictions on other major components in order to achieve a comprehensive evaluation of its drinking, medical and industrial value. Through the establishment of various intelligent models, we aim to provide the neces-sary data foundation for the smartification of tea gardens and IoT device systems. This will enable the early perception and prediction of soil conditions and related tea quality, allowing for early intervention to ensure the quality of tea leaves.
We would like also to thank you for allowing us to resubmit a revised copy of the manuscript.
We hope that the revised manuscript is accepted for publication in the Journal of Agriculture-Basel. Thanks again!
Kind regards,
Baijuan Wang
E-Mail: wangbaijuan2023@163.com

Reviewer 3 Report
Comments and Suggestions for Authors
This study is amied to predict flavonoids content in ancient tree sun dried green tea under abiotic stress based on machine learning. the study design is acceptable. The study contains some valuable results. Discussion is too short. The study needs many revisions before considering for possible publications.
Suggestions:
L18: Give in full 'LASSO'.
L26: AIC is not introduced yet.
L35-44: Long section without any citation.
L96: At the end of Introduction you need to give a one sentence well defined objectives.
M and M
This section is too short. You should give a more detailed information about the study area and also about the study design and analyses. Several M and M parts are included in the Results.
L168-171: This belongs to M and M.
L213-215: This belongs to M and M.
L244: Figure 2: Letter size is not large enough. The title is too short not well informative.
L258: Unreadable figures. Small letters. Title is to short and not informative.
L285: Figure 4 a and Figure 4 b are not explained. The Title is too short and not informative. Give in full FP, TP and ROC in the title.
L295: Figure 5 is unvisible. Figure 5a,b,c,d are not explained. The title is too short and not informative.
L339: Table 2. Title is not informative.
L340: Discussion is too short and give not deep enough comparison with previous results.
Author Response
Dear Reviewer,
Thanks very much for taking your time to review this manuscript. I really appreciate all your comments and suggestions! Please find my itemized responses in below and my revisions/corrections in the re-submitted files.
Here are the journal comments from the reviewer:
- L18: Give in full LASSO.
Thank you for your reminder. We have added the full name of “LASSO” to line 20 of the current manuscript.
- L26: AIC is not introduced yet.
I am thankful for your guidance, We have added the introduction of “AIC” to line 23-24 of the current manuscript.
- L35-44: Long section without any citation.
Thank you sincerely for your advice. We have added two additional references at line 48 of the current manuscript.
- L96: At the end of Introduction you need to give a one sentence well defined objectives
Thank you very much for your suggestion. This suggestion is crucial for the article structure. We have made the necessary revisions, and the changes can be found in lines 102-105 of the current manuscript.
- This section is too short. You should give a more detailed information about the study area and also about the study design and analyses. Several M and M parts are included in the Results
Thank you for your guidance. We have made the necessary revisions to the Materials and Methods section according to your recommendations. Here is the revised Materials and Methods section:
2.1. Tea and soil sample collection
This study selected eight main cultivation areas of ancient tea trees in Shahe Village, Hetou Village, and Tangfang Village of Laowu Mountain, Zhenyuan County, Pu’er City (N23° E100°) as sampling points. Samples were taken of the selected ancient tea trees for the production of sun-dried green tea and soil. Before excavating the soil of the roots of ancient tea tree, the surface litter and a 4-5cm layer of soil were removed. The five-point sampling method was used to collect samples from the soil within a vertical depth range of 20cm and 40cm. Samples were collected in triplicate at each sampling point, with each sample weighing no less than 200g. Uniform depth and weight were ensured at each sampling point. A total of 180 samples of ancient tree rhizosphere soil and 90 samples of ancient tree tea sun-dried green tea were collected for this study. Each sample was numbered, and sampling records and sample labels were filled out for each sample.
2.2. Detection methods
The detection method for total flavonoids in tea samples adopts the aluminum chloride colorimetric method. The determination of soil pH was carried out using the potentiometry, the determination of organic matter was obtained from the oxidation of potassium dichromate oxidation spectrophotometric determination of organic carbon multiplied by the constant 1.724, The determination of soil specific gravity is conducted using the pycnometer method, the determination of total phosphorus by alkali fu-sion-Mo-Sb anti spectrophotometric method, the determination of available phospho-rus by sodium hydrogen carbonate solution-Mo-Sb anti spectrophotometric method, The determination of total potassium content is conducted using the ICP-AES method, determination of available potassium content in soil using neutral ferric acetate solution leaching and flame photometry, the determination of total nitrogen by Kjeldahl method, the determination of ammonium nitrogen content is done using a universal ex-tract-colorimetric method, the determination of nitrate nitrogen content is done using a phenol-two-sulfonic acid method, the determination of hydrolytic nitrogen content involves hydrolyzing with a sodium hydroxide solution, adding a boric acid solution, and then using a standard acid titration method for determination.
2.3. Statistical analysis
This study utilized R language version 4.1.2 and used LASSO (Least Absolute Shrinkage and Selection Operator) regression to select modeling factors from 90 sam-ples of ancient tree tea sun-dried green tea and their corresponding samples of ancient tea tree rhizosphere soil [27,28]. The main components of this regression include re-ducing the number of variables, creating a penalty function, reducing the coefficients of variables, and forcing some regression coefficients to become zero. LASSO Regression is a biased partial method for handling data with multicollinearity, which also has the advantage of subset shrinkage. In addition, LASSO regression is effective in reducing model complexity and the number of required dependent variable types. By controlling parameters, it prevents overfitting [37,38]. In the feature selection phase of this study’s LASSO regression model, strongly correlated variables were selected as modeling fac-tors for machine learning model construction from age of tree, altitude, ammonium ni-trogen-20, ammonium nitrogen-40, available phosphorus-20, available phosphorus-40, exchangeable potassium-20, exchangeable potassium-40, hydrolytic nitrogen-20, hydrolytic nitrogen-40, nitrate nitrogen-20, nitrate nitrogen-40, organic carbon-20, organic carbon-40, organic matter-20, organic matter-40, pH-20, pH-40, specificgravity-20, specificgravity-40, total nitrogen-20, total nitrogen-40, total phosphorus-20, total phosphorus-40, total potassium-20, total potassium-40,tree height, which based on the analysis of the coefficient distribution and cross-validation results of LASSO. In order to confirm the correlation between modeling factors and flavonoid content, this research randomly split the dataset into a training set and a validation set, with a ratio of 8:2, then performed a single factor analysis by employing Cox regression on all variables [29,30]. Additionally, Cox regression also performed a multifactor analysis of the selected modeling factors. AIC, which measures the adequacy of statistical models based on entropy, serves as a metric for balancing the complexity of estimated models and the fit to the data. Typically, the model with the lowest AIC value is preferred when selecting the best model. So we also validated the modeling factors using the Akaike Information Criterion (AIC) [31,32].
This study utilized ROC curves and calibration curves to evaluate the accuracy and stability of the model. The ROC curve measures the performance of a classification model by calculating the area under the curve, while the calibration curve is used to compare the consistency between actual and predicted results [33,34].
In order to address the limitation of a small modeling dataset, this research additionally introduced the bootstrap method for dataset augmentation [35,36]. The fundamental concept of bootstrap was to obtain numerous samples from the initial dataset, conduct repeated experiments, and create multiple different datasets. And then use the empirical distribution of these data sets to substitute the population distribution. This is done by using the random put-back sampling method, where a certain number of samples are taken from the original dataset, and based on these samples the estimated statistic is calculated and the variance and distribution are estimated based on the re-sults of the calculations.
By introducing the bootstrap method, this research can leverage a larger dataset for modeling, thereby improving the evaluation of accuracy and stability of the model. This can increase model performance and the consistency of the results.
- L168-171: This belongs to M and M
Thank you for your advice. We have relocated this section to lines 167-169 of the Materials and Methods section in the current manuscript.
- L213-215: This belongs to M and M
Thank you so much for your suggestion. We have relocated this section to lines 184-186 of the Materials and Methods section in the current manuscript.
- L244: Figure 2: Letter size is not large enough. The title is too short not well informative.
Thank you very much for your valuable feedback. We have made changes to Figure 2 of the previous manuscript and provided more detailed annotations for the title. The changed figure title is: Nomogram model predicts the relationship between the range of variation of six strongly correlated factors and their corresponding flavonoids content, and the modified content is in line 324-326.
- L258: Unreadable figures. Small letters. Title is to short and not informative.
Thank you for your suggestion. We have made modifications to Figure 3 (now Figure 4) and provided a detailed explanation in the title. The changed figure title is: Calibration curve. 4a shows the calibration curve of the training set when the flavonoids content is >6‰, 4b shows the calibration curve of the training set when the flavonoids content is >9‰, 4c shows the calibration curve of the validation set when the flavonoids content is >6‰, 4d shows the calibration curve of the validation set when the flavonoids content is >9‰. And the modified content is in line 349-352.
- L285: Figure 4 a and Figure 4 b are not explained. The Title is too short and not informative. Givein full FP TP and ROC in the title.
Thank you very much for your valuable advice. We have already explained Figures 4a and 4b (now Figure 5a and 5b) and provided a detailed caption. The changed figure title is: ROC curve analysis. When the flavonoids content is above 6‰ in this model, the AUC values achieved for the training set and validation set were 0.8121 and 0.792, respectively. However, if the flavonoids content surpasses 9‰, the AUC values for the training set and validation set im-prove to 0.877 and 0.889, respectively. The horizontal axis FP represents False Positive, and the vertical axis TP represents True Positive. And the modified content is in line 382-386.
- L295: Figure 5 is unvisible. Figure 5a, b, c, d are not explained. The title is too short and notinformative.
Thank you very much for your valuable feedback. We have made changes to Figure 5 and provided explanations for Figures 5a, 5b, 5c, and 5d, as well as detailed the information in the captions, the modified content is: Main interface of the visualization system for flavonoids content prediction. 6a is survival plot module,6b is predicted survival module,6c is numerical summary module,6d is model summary module, which now in line 397-399.
- Table 2. Title is not informative
Thank you for your advice. We have provided a more detailed explanation for the original Table 2 in our manuscript,the modified content is: External validation results of the flavonoids prediction model using 12 sets of data, as mentioned in line 454-455.
- L340: Discussion is too short and give not deep enough comparison with previous results
Thank you for your reminder. We have made changes to the discussion section and added limitations, future research directions, and the importance of our study. Below is the revised discussion section we have made:
At present, traditional line graph models have problems such as inability to analyze and predict data quickly, and to display multiple results simultaneously. In this study, a LASSO-Cox regression model was used to select six significant variables for con-structing a column-line chart model. This model can provide data and theoretical sup-port for soil management in the Laowu Mountain ancient tea trees garden, it can also predict the content of catechins in the ancient tree sun-dried green tea of Laowu Mountain based on soil conditions, 3while ensuring balanced taste, the flavonoids content can be increased appropriately [45], thereby enhancing the antioxidant [46], immune regulation [47], anti-cancer [48], anti-inflammatory, detoxifying effects of an-cient tea. However, due to the concentrated collection of specimens in the ancient tea gardens of Laowu Mountain during the spring season, there is a lack of sample data from other seasons and regions. Yunnan has a vast and diverse tea-growing region, with each tea-growing area having its own microclimate and soil conditions. Therefore, it is crucial to supplement the data on soil conditions and flavonoid content from other production regions. The next step for the team is to collect data on different seasons of Laowu Mountain and other sun-dried green tea producing regions, in order to improve the data system and establish a more universal predictive model. Tea contains a variety of substances, and they collectively determine the taste and effectiveness of tea. In the future, our team will also make predictions on other major components in order to achieve a comprehensive evaluation of its drinking, medical and industrial value. Through the establishment of various intelligent models, we aim to provide the neces-sary data foundation for the smartification of tea gardens and IoT device systems. This will enable the early perception and prediction of soil conditions and related tea quality, allowing for early intervention to ensure the quality of tea leaves.
Due to the large number of files, we have consolidated them into a folder and uploaded them in the form of a compressed file. We would like also to thank you for allowing us to resubmit a revised copy of the manuscript.
We hope that the revised manuscript is accepted for publication in the Journal of Agriculture-Basel. Thanks again!
Kind regards,
Baijuan Wang
E-Mail: wangbaijuan2023@163.com

Reviewer 4 Report
Comments and Suggestions for Authors
The study attempted to predicted flavonoid percentage in ancient tree tea using some sort of “machine learning” algorithm, which I did not get a clear answer of after reading the manuscript.
The manuscript was, arguably intentionally, vaguely written, missing important details throughout.
In introduction, no knowledge gaps and justifications were provided.
In materials and methods, sampling point location is missing. Line 99-109 is simply insufficient for explaining data collection and photo evidence and explanation would be necessary. Line 110-125 lacks necessary detailed description and explanation.
Line 135-139, 181-184, input variable list into the LASSO regression is missing. I know it’s in Table 1, yet without supplying the information disrupts the content flow.
Section 3.2, selecting variable using LASSO first, and then go back and analyze all variables using Cox is simply redundant. Nonetheless, Table 1 needs to be completed instead of having empty cells.
Line 227 what machine learning model? Why it was not even mentioned in methods?
Figure 2-5, unprofessional titles.
In my opinion, the authors intentionally deployed all sorts of “fancy” yet unimportant tricks to distract readers from the insignificant results obtained in the study, such as LASSO regression, Cox regression, AIC values, bootstrap, mysterious “machine learning” “points”, orange “calibration curve”, ROC curve, and finally a “visualization system”, turning a simple regression problem of tea flavonoid into a lengthy, complex, and empty evaluation report of the insignificant results obtained in Figure 3 even after using bootstrap. To further complicate the problem and disguise the insignificant results, in the last result section, the authors turned the simple regression problem into a classification problem based on their developed “visualization system” to claim the “83.33% accuracy”. The results of the study can be simply evaluated by a scatter plot between prediction and observation, using R2 and RMSE as evaluation metrics.
Based on the study results, the statement “through soil prediction, the flavonoids content of ancient trees can be predicted.” in line 341 is not true. The discussion section is the shortest discussion I’ve ever read by the way.
Author Response
Dear Reviewer,
Thanks very much for taking your time to review this manuscript. I really appreciate all your comments and suggestions! Please find my itemized responses in below and my revisions/corrections in the re-submitted files.
Here are the journal comments from the reviewer:
- The manuscript was, arguably intentionally, vaguely written, missing important details throughout.
Thank you for your reminder. We have made more refined and elaborate writing, providing more detailed descriptions for the entire manuscript including abstract, introduction, materials and methods, results, discussion, and conclusion.
- In introduction, no knowledge gaps and justifications were provided.
We are thankful for your guidance. We have revised and improved the introduction.
- In materials and methods, sampling point location is missing. Line 99-109 is simply insufficient for explaining data collection and photo evidence and explanation would be necessary. Line 110-125 lacks necessary detailed description and explanation.
Thank you for your reminder. In the manuscript, we have supplemented the position of the sampling point with words (in line 109-111) and coordinates with pictures (in line 122-123).
In the original manuscript, we described the measurement methods of 12 soil factors in lines 110-125. The first method is the determination of soil pH, which can be roughly tested as follows: using water as the leaching agent and a water-soil ratio of 2.5:1, the indicator electrode and reference electrode (or pH composite electrode) are immersed in the soil suspension, forming a primary cell. At a certain temperature, the electric potential of the primary cell and the pH value of the suspension are determined. By analyzing the electric potential of the primary cell, we can obtain the pH value of the soil. If we were to provide detailed descriptions of the methods, calculation formulas, reagents, and instruments used for measuring the 12 soil factors, it would not only occupy too much space, but also deviate from the research focus of this manuscript.
- Line 135-139, 181-184, input variable list into the LASSO regression is missing. I know it’s in Table 1, yet without supplying the information disrupts the content flow.
Thank you for your reminder. According to your suggestion, we have added all the measured variables in line 169-178 and 229-233 of the revised manuscript.
- Section 3.2, selecting variable using LASSO first, and then go back and analyze all variables using Cox is simply redundant. Nonetheless, Table 1 needs to be completed instead of having empty cells.
Thank you very much for your advice. The reason why Cox regression was used for further analysis in this study is to further improve the accuracy of the prediction model by analyzing and selecting the modeling factors filtered by LASSO. This approach can also reduce unnecessary workload during the model establishment process. Additionally, research by Wang et al. has shown that combining LASSO with Cox can improve the accuracy of prediction models. We have made modifications to Table 1 in the original manuscript.
Reference: Wei Wang, Wei Liu, PCLasso: a protein complex-based, group lasso-Cox model for accurate prognosis and risk protein complex discovery, Briefings in Bioinformatics, Volume 22, Issue 6,November 2021, bbab212, https://doi.org/10.1093/bib/bbab212.
- Line 227 what machine learning model? Why it was not even mentioned in methods?
Thank you for your reminder. We have altered and explained in detail the content of the original text, which is presented in lines 290-326 of the altered manuscript.
- Figure 2-5, unprofessional titles.
We are thankful for your guidance, we have now made changes to the figure title, which are as follows:
Figure S1: Sampling point distribution map;
Figure S2: Factor Screening Based on LASSO Regression: Figure 2a displays the distribution of coefficients in LASSO regression, and Figure 2b presents the cross-validation plot;
Figure S3: Nomogram model predicts the relationship between the range of variation of six strongly correlated factors and their corresponding flavonoids content;
Figure S4: Calibration curve. 4a shows the calibration curve of the training set when the flavonoids content is >6‰, 4b shows the calibration curve of the training set when the flavonoids content is >9‰, 4c shows the calibration curve of the validation set when the flavonoids content is >6‰, 4d shows the calibration curve of the validation set when the flavonoids content is >9‰;
Figure S5: ROC curve analysis. When the flavonoids content is above 6‰ in this model, the AUC values achieved for the training set and validation set were 0.8121 and 0.792, respectively. However, if the flavonoids content surpasses 9‰, the AUC values for the training set and validation set improve to 0.877 and 0.889, respectively. The horizontal axis FP represents False Positive, and the vertical axis TP represents True Positive;
Figure S6: Main interface of the visualization system for flavonoids content prediction. 6a is survival plot module,6b is predicted survival module,6c is numerical summary module,6d is model summary module.
- In my opinion, the authors intentionally deployed all sorts of “fancy” yet unimportant tricks to distract readers from the insignificant results obtained in the study, such as LASSO regression, Cox regression, AIC values, bootstrap, mysterious “machine learning” “points”, orange “calibration curve”, ROC curve, and finally a “visualization system”, turning a simple regression problem of tea flavonoid into a lengthy, complex, and empty evaluation report of the insignificant results obtained in Figure 3 even after using bootstrap. To further complicate the problem and disguise the insignificant results, in the last result section, the authors turned the simple regression problem into a classification problem based on their developed “visualization system” to claim the “83.33% accuracy”. The results of the study can be simply evaluated by a scatter plot between prediction and observation, using R2 and RMSE as evaluation metrics.
Thank you very much for your advice and guidance. Compared to traditional research in the past, this study requires the establishment of a prediction model, therefore, more complex techniques and comprehensive model evaluation are needed to improve and validate the accuracy of the model predictions. In this study, LASSO was used for the preliminary screening of predictors for flavonoids prediction modeling. In order to further improve the accuracy of the prediction, we further selected factors strongly correlated with flavonoids content using Cox regression. This step is necessary not only for the accuracy of the later predictions, but also to reduce unnecessary workload when establishing the model. After the above screening, AIC was introduced to validate the selected predictors, and the strongly correlated factors that have been validated were visualized using nomogram model to show their positive or negative correlation and strong or weak correlation. This provides a more accurate basis for subsequent data analysis and tea garden soil management. To evaluate the established model, ROC curves and calibration curves were used in this study. Additionally, Bootstrap was used to increase the sample size of the model and increase the accuracy of the prediction results while reducing prediction bias caused by small sample size. The research methodology used in this study is also in line with that used in the medical field regarding machine learning by e.g. Zhou, Wang et al. These ideas regarding the research methodology, combined with your valuable and helpful suggestions, have led us to make revisions in various sections of the paper, including the introduction, methods, results, discussion, and conclusion. Once again, we sincerely thank you for your advice.
References: Zhou, D., Liu, X., Wang, X. et al. A prognostic nomogram based on LASSO Cox regression in patients with alpha-fetoprotein-negative hepatocellular carcinoma following non-surgical therapy. BMC Cancer 21, 246 (2021). https://doi.org/10.1186/s12885-021-07916-3.
Qi Wang, Wenying Qiao, Honghai Zhang, et al. Nomogram established on account of Lasso-Cox regression for predicting recurrence in patients with early-stage hepatocellular carcinoma. Frontiers in immunology, November 2022. https://doi.org/10.3389/fimmu.2022.1019638.
- Based on the study results, the statement “through soil prediction, the flavonoids content of ancient trees can be predicted.” in line 341 is not true. The discussion section is the shortest discussion I’ve ever read by the way.
We are thankful for your guidance. We have changed the entire conclusion and checked it sentence by sentence, and here is our changed conclusion:
At present, traditional line graph models have problems such as inability to analyze and predict data quickly, and to display multiple results simultaneously. In this study, a LASSO-Cox regression model was used to select six significant variables for con-structing a column-line chart model. This model can provide data and theoretical sup-port for soil management in the Laowu Mountain ancient tea trees garden, it can also predict the content of catechins in the ancient tree sun-dried green tea of Laowu Mountain based on soil conditions, 3while ensuring balanced taste, the flavonoids content can be increased appropriately [43], thereby enhancing the antioxidant [44], immune regulation [45], anti-cancer [46], anti-inflammatory, detoxifying effects of an-cient tea. However, due to the concentrated collection of specimens in the ancient tea gardens of Laowu Mountain during the spring season, there is a lack of sample data from other seasons and regions. Yunnan has a vast and diverse tea-growing region, with each tea-growing area having its own microclimate and soil conditions. Therefore, it is crucial to supplement the data on soil conditions and flavonoid content from other production regions. The next step for the team is to collect data on different seasons of Laowu Mountain and other sun-dried green tea producing regions, in order to improve the data system and establish a more universal predictive model. Tea contains a variety of substances, and they collectively determine the taste and effectiveness of tea. In the future, our team will also make predictions on other major components in order to achieve a comprehensive evaluation of its drinking, medical and industrial value. Through the establishment of various intelligent models, we aim to provide the necessary data foundation for the smartification of tea gardens and IoT device systems. This will enable the early perception and prediction of soil conditions and related tea quality, allowing for early intervention to ensure the quality of tea leaves.
We would like also to thank you for allowing us to resubmit a revised copy of the manuscript.
We hope that the revised manuscript is accepted for publication in the Journal of Agriculture-Basel. Thanks again!
Kind regards,
Baijuan Wang
E-Mail: wangbaijuan2023@163.com

Round 2
Reviewer 3 Report
Comments and Suggestions for Authors
The study improved. It can be considered for publication.
Author Response
Dear Reviewer,
Thank you for dedicating your time to reviewing this manuscript. We are extremely grateful for your valuable feedback, which has significantly improved the content and overall organization of our work. We believe that everything will develop towards a better direction.
Kind regards,
Baijuan Wang
E-Mail: wangbaijuan2023@163.com
Reviewer 4 Report
Comments and Suggestions for Authors
I appreciate the authors’ efforts in improving the manuscript quality. However, I do not think the authors have addressed some of my comments sufficiently.
To my comment on vague method description in line 110-125, the authors responded “If we were to provide detailed descriptions of the methods, calculation formulas, reagents, and instruments used for measuring the 12 soil factors, it would not only occupy too much space, but also deviate from the research focus of this manuscript.” I would like to remind the authors that the purpose of providing detailed method descriptions is to allow other researchers to replicate the results of the study. Minimally, the authors should provide necessary references for each method that explains the method in detail.
I insist the authors to provide one-to-one scatter plots of predicted and observed flavonoids contents in the same unit, using R2 and RMSE as evaluation metrics, and modify the study conclusions accordingly. I will reevaluate the manuscript after see this result.
Author Response
Dear Reviewer,
Thanks very much for taking your time to review this manuscript. I really appreciate all your comments and suggestions! Please find my itemized responses in below and my revisions/corrections in the re-submitted files.
Here are the journal comments from the reviewer:
- To my comment on vague method description in line 110-125, the authors responded “If we were to provide detailed descriptions of the methods, calculation formulas, reagents, and instruments used for measuring the 12 soil factors, it would not only occupy too much space, but also deviate from the research focus of this manuscript.” I would like to remind the authors that the purpose of providing detailed method descriptions is to allow other researchers to replicate the results of the study. Minimally, the authors should provide necessary references for each method that explains the method in detail.
We are thankful for your guidance. The focus of this research is to establish a prediction model for the range of total flavonoids content based on the relationship between soil mineral content and total flavonoids. The purpose of this study is to provide a basis and support for the intelligentization of the tea industry, IoT devices, and systems by establishing the model, and to achieve tea quality prediction and intervention. Therefore, this study provides a detailed description of the method for model establishment, so that other researchers can establish corresponding predictive models using relevant data. As the methods section would be too long if we were to describe the 12 soil factors detection methods, we have summarized the total flavonoids and soil detection methods used in this study. The methods are as follows:
The first method is the determination of total flavonoids, which can be roughly tested as follows: after the interaction between flavonoids compounds in tea leaves and AlCl3, aluminum complexes of flavonoids are formed, which are yellow in color. The intensity of the color is proportional to the flavonoid content. By comparing with a standard curve, quantitative analysis can be conducted. Most of the flavonoid substances in tea leaves exist in the form of glycosides, so flavonoid glycosides can be used as the reference substance for establishing a quantitative standard curve.
The second method is the determination of soil pH, which can be roughly tested as follows: using water as the leaching agent and a water-soil ratio of 2.5:1, the indicator electrode and reference electrode (or pH composite electrode) are immersed in the soil suspension, forming a primary cell. At a certain temperature, the electric potential of the primary cell and the pH value of the suspension are determined. By analyzing the electric potential of the primary cell, we can obtain the pH value of the soil.
The third method is the determination of organic matter, which can be roughly tested as follows: under heating conditions, excess potassium dichromate solution is used to oxidize soil organic carbon, and the excess potassium dichromate is titrated with ferrous sulfate standard solution. The amount of consumed potassium dichromate is calculated based on the oxidation correction factor to determine the amount of organic carbon, which is then multiplied by a constant of 1.724 to obtain the soil organic matter content.
The fourth method is the determination of organic carbon, which can be roughly tested as follows: under heating conditions, organic carbon in the soil sample is oxidized by an excess of potassium dichromate-sulfuric acid solution. The hexavalent chromium (Cr6+) in the potassium dichromate is reduced to trivalent chromium (Cr3+), and its content is directly proportional to the organic carbon content in the sample. The absorbance is measured at a wavelength of 585 nm, and the organic carbon content is calculated based on the content of trivalent chromium (Cr3+).
The fifth method is the determination of soil specific gravity, which can be roughly tested as follows: the bulk density of soil is the ratio of the mass of the solid portion of the soil to the mass of an equal volume of water at a temperature of 40°C. The density bottle method is commonly used to determine the bulk density of soil, using the principle of the drainage weighing method to measure the mass of water in the same volume. The water content of the soil is then measured to obtain the mass of dry soil (at 105°C), which is divided by the volume to obtain the bulk density. The temperature has an effect on the density of water, so it is necessary to ensure consistent temperature during the weighing before and after testing. For soils with a high content of soluble salts or active colloids, a non-polar liquid is used instead of water, and the sample needs to be pre-dried to a constant mass. Vacuum extraction is used instead of boiling to remove air from the soil.
The sixth method is the determination of total phosphorus, which can be roughly tested as follows: after being melted with sodium hydroxide, the phosphorus-containing minerals and organic phosphorus compounds in the soil samples are converted into soluble orthophosphates. Under acidic conditions, they react with molybdate-antimony color reagents to form phosphomolybdenum blue, which is measured for absorbance at a wavelength of 700 nm. Within a certain concentration range, the total phosphorus content in the samples follows the Lambert-Beer law in relation to the absorbance values.
The seventh method is the determination of available phosphorus, which can be roughly tested as follows: the available phosphorus in a 0.5 mol/L sodium bicarbonate solution (pH 8.5) was extracted. The extracted phosphate reacted with a molybdoantimony reagent to form phosphomolybdenum blue, which was measured for absorbance at a wavelength of 880 nm. Within a certain concentration range, the phosphate content and absorbance values followed the Lambert-Beer law.
The eighth method is the determination of total potassium, which can be roughly tested as follows: dissolve 1.9067g of KCl (burned at 400~450°C without explosion) in water and make up to 1L with water to prepare a standard stock solution of 1.00mg/ml. After calibrating the soil sample solution prepared according to the instructions for instrument use, perform sample measurements in the optimal working parameter state of the instrument. Subtract background or correct interference using the interference coefficient method.
The ninth method is the determination of available potassium, which can be roughly tested as follows: available potassium is extracted with neutral 1 mol/L acetic acid solution and determined by flame photometry.
The tenth method is the determination of total nitrogen, which can be roughly tested as follows: the total nitrogen in the soil is completely converted to nitrate nitrogen through oxidation-reduction reactions under the action of sodium thiosulfate, concentrated sulfuric acid, perchloric acid, and catalyst. The ammonia distilled by alkalizing the digested solution is absorbed by boric acid and titrated with standard hydrochloric acid solution. The amount of standard hydrochloric acid solution is used to calculate the total nitrogen content in the soil.
The eleventh method is the determination of ammonium nitrogen, which can be roughly tested as follows: the Na+ in the joint extracting agent can exchange with NH+ and K+ on the surface of soil colloids, together with water-soluble ions, entering the solution. In acidic soil, phosphorus mainly exists in the form of Fe-P and Al-P. The ability of Fe3+ and Al3+ to complex with F in acidic solution causes a certain amount of active phosphorus to be released from phosphorus iron and phosphorus aluminum, and at the same time, the phosphorus in Ca-P with larger activity is also dissolved out due to the action of H+. The ammonium ions in the leachate react with the sodium reagent to generate a yellow substance. Within a certain concentration range, the depth of its color is directly proportional to the content of nitrate nitrogen in the solution, as measured at a wavelength of 420nm.
The twelfth method is the determination of nitrate nitrogen, which can be roughly tested as follows: After extracting the soil sample with a saturated solution of calcium sulfate, a portion of the extract is evaporated to dryness under weak alkaline conditions. The residue is then treated with phenol disulfonic acid. At this point, nitrate nitrogen reacts with phenol disulfonic acid to produce nitrophenol disulfonic acid. This reaction can only be rapidly completed under anhydrous conditions. The reaction product is colorless in acidic medium, but turns into a stable yellow salt solution after alkalization. The color can be measured by colorimetry at a wavelength of 420 nm.
The thirteenth method is the determination of hydrolytic nitrogen, which can be roughly tested as follows: treat the soil with a 1.8 mol/L sodium hydroxide solution. In a diffusion dish, the soil undergoes hydrolysis under alkaline conditions, converting readily available nitrogen into ammonium nitrogen through alkaline hydrolysis. After diffusion, the nitrogen is absorbed by a boric acid solution and is calculated by standard acid titration to determine the content of hydrolyzed nitrogen. If the forest soil has a high content of nitrate nitrogen, a reducing agent should be added for reduction. However, for the initial soil of the forest soil, since the nitrate nitrogen content is low, there is no need to add a reducing agent for reduction. Therefore, the concentration of sodium hydroxide solution can be reduced to 1.2 mol/L.
- I insist the authors to provide one-to-one scatter plots of predicted and observed flavonoids contents in the same unit, using R2 and RMSE as evaluation metrics, and modify the study conclusions accordingly. I will reevaluate the manuscript after see this result.
Thank you very much for your advice. Due to the fact that this study aims to predict the total flavonoid content within 3 different ranges based on the synergistic effect of 6 strongly correlated factors, rather than predicting a specific value of total flavonoids content, a calibration curve was employed in the methodology of this study to evaluate and validate model performance. This approach not only helps us understand the accuracy of the model within different prediction probability ranges, but also allows targeted adjustments and improvements to the model. We appreciate your suggestion, and we tried using R2 and RMSE, but found that neither of these methods can achieve a fit analysis for interval prediction. The total flavonoids content in tea plants is influenced by multiple factors working synergistically, such as varying soil pH, even if the hydrolytic nitrogen content, which has the most significant impact on total flavonoids content, remains consistent, the total flavonoids content can still differ. Moreover, in this study, the soil samples are from ancient tea gardens, which do not allow for the specific impact analysis of individual variables on flavonoids content. Nevertheless, we are grateful for your suggestion as it provides a valuable direction for our future research. In the next step, we can conduct pot experiments, where other relevant factors are kept constant, and examine the one-to-one influence of soil factors on flavonoid content by controlling the variation of individual variables.
We would like also to thank you for allowing us to resubmit a revised copy of the manuscript.
We hope that the revised manuscript is accepted for publication in the Journal of Agriculture-Basel. Thanks again!
Best regards,
Wang Baijuan
